# Influence of Different Salts on the G-Quadruplex Structure Formed from the Reversed Human Telomeric DNA Sequence

**DOI:** 10.3390/ijms232012206

**Published:** 2022-10-13

**Authors:** Lydia Olejko, Anushree Dutta, Kosar Shahsavar, Ilko Bald

**Affiliations:** Institute of Chemistry–Hybrid Nanostructures, University of Potsdam, 14476 Potsdam, Germany

**Keywords:** guanine quadruplexes, förster resonance energy transfer (FRET), circular dichroism, DNA nanotechnology

## Abstract

G-rich telomeric DNA plays a major role in the stabilization of chromosomes and can fold into a plethora of different G-quadruplex structures in the presence of mono- and divalent cations. The reversed human telomeric DNA sequence (5′-(GGG ATT)_4_; RevHumTel) was previously shown to have interesting properties that can be exploited for chemical sensing and as a chemical switch in DNA nanotechnology. Here, we analyze the specific G-quadruplex structures formed by RevHumTel in the presence of K^+^, Na^+^, Mg^2+^ and Ca^2+^ cations using circular dichroism spectroscopy (CDS) and Förster resonance energy transfer (FRET) based on fluorescence lifetimes. CDS is able to reveal strand and loop orientations, whereas FRET gives information about the distances between the 5′-end and the 3′-end, and also, the number of G-quadruplex species formed. Based on this combined information we derived specific G-quadruplex structures formed from RevHumTel, i.e., a chair-type and a hybrid-type G-quadruplex structure formed in presence of K^+^, whereas Na^+^ induces the formation of up to three different G-quadruplexes (a basket-type, a propeller-type and a hybrid-type structure). In the presence of Mg^2+^ and Ca^2+^ two different parallel G-quadruplexes are formed (one of which is a propeller-type structure). This study will support the fundamental understanding of the G-quadruplex formation in different environments and a rational design of G-quadruplex-based applications in sensing and nanotechnology.

## 1. Introduction

Apart from the well-known double helix structure, DNA can also adopt fascinating higher order structures such as guanine (G) quadruplex or i-motif structures [1,2,3,4]. Guanine-rich DNA sequences, usually found in telomeres (located at the end of eukaryotic chromosomes) [5,6], tend to fold in the presence of metal ions to form secondary structures of two or more G-tetrads termed as G-quadruplex. The G-tetrad plane consists of four associated G bases stabilized by eight Hoogsteen hydrogen bonds and a cation in the middle of the G-tetrad (see Figure 1A) [4,7,8,9]. The metal cation coordination to the four central oxygen atoms of the G bases weakens the electronic repulsion between the oxygen atoms resulting in stronger hydrogen bonds and therefore stabilization of the G-tetrad [10]. Based on the number of DNA strands (four, two and one) involved in the folding, G-quadruplexes can be classified into tetramolecular, bimolecular or intermolecular and unimolecular or intramolecular, respectively. Similarly, depending on the DNA strand orientation, the G-quadruplex structure can be divided into anti-parallel (opposite strand orientations) and parallel (same strand orientation) structures (see Figure 1B) [4,9,10,11]. The G-tetrads can be connected with lateral and diagonal loops leading to so-called “basket-type” or “chair-type” G-quadruplexes (anti-parallel) or with external loops leading to “propeller-type” G-quadruplexes (parallel) (see Figure 1B). A combination of lateral and/or diagonal loops with external loops lead to “hybrid-type” G-quadruplexes (combination of anti-parallel and parallel, Figure 1B) [4,9,10]. Therefore, based on different loop and strand orientations, the distance between 5′- and 3′-end will differ in the G-quadruplex structure varying from 4 nm to 6 nm, as schematically depicted in Figure 1B,C. Herein, the distances between 5′- and 3′-end in the folded structure have been calculated based on the X-ray crystallography, NMR and molecular dynamics data reported in literature for the human telomeric DNA [12,13].

CD spectroscopy, which is based on the effect that chiral (optically active) molecules absorb right- and left-handed circularly polarized light differently and can be used for structural investigations of biomolecules such as DNA [14]. The same has been extended to determine the specific structure of G-quadruplexes using circular dichroism (CD) spectroscopy [15,16]. In this sense, it can also be used to analyze the different structures of certain G-quadruplexes which ideally should lead to different CD spectra. This effect is ascribed to the varying stacking interactions due to different glycosidic angles and strand orientations [17,18]. For example, a parallel G-quadruplex structure should give rise to a positive peak at (260–265) nm and a negative peak at 240 nm. Positive peaks at (290–295) nm and (240–250) nm and a negative peak at (240–275) nm are a sign of the formation of anti-parallel G-quadruplex structures. On the other hand, hybrid-type G-quadruplex structures lead to a positive peak at 290 nm with a shoulder at (260–270) nm and a negative peak at 240 nm [15,16,19,20,21,22,23].

From the human telomeric DNA (HumTel, (TTA GGG)_n_), it is known that environmental conditions have a great influence on the overall G-quadruplex structure (see also Appendix A). Many studies revealed that a variety of changes both within the DNA sequence and external conditions influence the formation of specific G-quadruplex structures [24,25]. The G-quadruplex structure is different for telomeric DNA and telomeric RNA [26], it is altered by changing the number of flanking nucleotides (AG_3_ (T_2_AG_3_)_3_, TAG_3_ (T_2_AG_3_)_3_, TAG_3_ (T_2_AG_3_)_3_T_2_ and A_3_G_3_ (T_2_AG_3_)_3_A_2_) [27,28], the number of involved repeating units ((TTA GGG)_2_ vs. (TTA GGG)_4_) [29,30] or the strand polarization (HumTel: (TTA GGG)_4_ vs. RevHumTel: (GGG ATT)_4_) [30]. When experimental conditions such as the cation itself (Na^+^, K^+^ and divalent cations) [12,31,32], the cation concentration [30,33,34], the DNA concentration [27,30], crowding agents (addition of polyethylene glycol to mimic molecular crowding in cells) [20] or the temperature [35,36] are changed, alterations of the G-quadruplex structure are expected [9]. Recently, it has been demonstrated that specific telomeric DNA sequences can be exploited in sensing because, e.g., the RevHumTel sequence formation of G-quadruplexes is highly ion-specific when connected to DNA nanostructures [30,36]. In this line, Noer et al. studied the folding dynamics and conformational heterogeneity exhibited by human telomeric G-quadruplex structures in the presence of Na^+^ ions by single-molecule FRET microscopy [37]. Therefore, in the present work, we took the time to analyze the influence of different salts of monovalent and divalent cations on the specific G-quadruplex structure and conformation of the reversed human telomeric DNA (RevHumTel, 5′-TT (GGG ATT)_4_) using CD spectroscopy and the principle of Förster resonance energy transfer (FRET). For this, we compare our measured CD spectra with previously published CD spectra of related G-quadruplex structures to comment on the different conformation of the G-quadruplex structure. To further support the CD spectroscopy results, FRET measurements were additionally performed to determine the distance between the 5′- and 3′-end (when modified with donor and acceptor molecules) and deduce the specific G-quadruplex structures [22,24,35]. Therefore, we considered the folding of unmodified RevHumTel in the presence of KCl, NaCl, MgCl_2_ and CaCl_2_ (0–500 mM). For the FRET experiments, the telomeric DNA has been modified with Cyanine3 (Cy3) at its 5′-end and with Fluorescein (FAM) at its 3′-end (5′-Cy3-TT (GGG ATT)_4_-FAM). Here, FAM acts as the FRET donor and Cy3 as the FRET acceptor. By combining the findings of CD spectroscopy and FRET experiments, we are able to estimate the specific G-quadruplex structure formed from RevHumTel in presence of different salts.

## 2. Results and Discussion

### 2.1. CD Spectroscopy

The CD spectra for RevHumTel (c = 8.6 µM, diluted in TAE buffer (1x)) for the different salts are shown in Figure 1. Different salts lead to different CD spectra. The telomeric DNA in a pure buffer (TAE 1x) gives rise to a negative peak at 240 nm and two positive peaks at 255 nm and 280 nm. The TAE buffer does not influence the CD spectra compared to the free telomeric DNA in ultra-pure water, as shown in the Appendix A. The CD spectrum changes when KCl is added to the solution (Figure 1A). After the addition of ca. 10 mM KCl, an intense positive peak at 288 nm with a shoulder at 275 nm becomes visible indicating the G-quadruplex formation. A broad negative CD band at (240–250) nm is also visible. Please note that the shape of the CD spectrum does not change significantly with an increasing amount of KCl, meaning that the salt concentration does not influence the specific G-quadruplex structure. Only the intensity of the CD bands rises with an increasing KCl concentration showing that the fraction of folded G-quadruplex structures increases. By comparing with CD spectra found in literature, the present CD spectra indicate the formation of a “hybrid-type” G-quadruplex structure by RevHumTel in presence of KCl [16].

The CD spectra after the NaCl addition (Figure 1B) look different compared to the CD spectra in the presence of KCl. More NaCl is needed to induce a change in the CD spectra, meaning that the G-quadruplex folds at higher NaCl concentrations (ca. 200 mM) compared to KCl (ca. 10 mM). This can be attributed to two different association constants in presence of KCl and NaCl (the association constant of KCl is higher compared to NaCl) [36]. Furthermore, the shape of the CD spectra after the G-quadruplex formation is rather different. In the presence of NaCl, the CD spectra have a minimum of 235 nm (not negative values) and two intense positive peaks at 270 nm and 290 nm. Again, the salt concentration does not influence the shape of the CD spectrum. Hence, the G-quadruplex structure is not dependent on the salt concentration. By comparing this CD spectrum with the previously reported CD spectra, the CD spectrum could be attributed to a “hybrid-type” or a mixture of parallel and anti-parallel G-quadruplexes [16].

It is not only the monovalent cations that can induce the formation of G-quadruplex structures. Divalent cations such as Mg^2+^ and Ca^2+^ are also known to stabilize G-quadruplexes [32,38,39], which is particularly important for G-quadruplexes connected to Mg^2+^-stabilized DNA nanostructures [30,36,40]. For MgCl_2_, the CD spectra are shown in Figure 1C. Here, the CD spectra can be clearly associated with a parallel G-quadruplex structure [15,16,19,20,21,22,23]. In presence of MgCl_2_, two intense signals are present (positive peak at (240–245) nm and negative peak at 265 nm). The concentration of MgCl_2_ needed for G-quadruplex formation is as low as 2 mM. Again, the shape of the CD spectra is almost unaffected by the salt concentration (which also accords with the concentration used in previous studies) [41,42] meaning that the G-quadruplex structure is also unaffected by the salt concentration [16].

The CD spectra for CaCl_2_ addition are shown in Figure 1D. Again, the CD spectrum changes at a salt concentration of 2 mM, meaning that the G-quadruplex structure folds at low CaCl_2_ concentrations. After the CaCl_2_ addition, a positive peak at 265 nm and a negative peak at 242 nm appears. This can also be attributed to a parallel G-quadruplex structure [15,16,19,20,21,22,23]. Please note that at high salt concentrations (500 mM), the peak at 265 nm decreases, which could indicate a partial destabilization of the G-quadruplex, which could occur by the accumulation of excess counterions in the vicinity of the G-quadruplex structure [43].

Finally, we have recorded CD spectra of RevHumTel in the presence of a mixture of ions corresponding to physiological conditions (i.e., 143 mM Na^+^, 5 mM K^+^, 2,5 mM Ca^2+^, 1 mM Mg^2+^, Appendix A). The CD spectrum shows a negative dip in the range 240–250 nm and a positive peak at around 290 nm with a shoulder at 270 nm indicating that in the mixture of salts under physiological conditions K^+^ ions dominate the formation of G-quadruplex structures, i.e., a hybrid-type G-quadruplex structure is formed. This is in line with melting curves measured by UV absorption at a concentration of 200 mM of each salt indicating the highest melting points for K^+^ induced G quadruplexes showing two melting stages (50 °C and 80 °C, Appendix A), whereas Na^+^ induced G-quadruplexes show the lowest melting point (23 °C).

### 2.2. FRET Experiments

To verify the findings of the CD spectroscopy, especially for ambiguous spectra we performed FRET experiments at constant salt concentrations to determine the 5′-3′-end distances (5′-end: Cy3, 3′-end: FAM). Please note that MgCl_2_ and CaCl_2_ are able to quench fluorescence at high concentrations. This could be attributed to dynamic quenching experienced by donor-acceptor dye pair(s) induced by divalent ions at higher concentrations [44]. Hence, measurements have been performed at concentrations as low as 30 mM and 2 mM, respectively. For KCl, the concentration has been set to 195 mM and for NaCl it has been set to 490 mM to assure a complete folding. The FRET efficiency [45,46]. E depends on the donor-acceptor distance R and behaves according to the following Equation (1),
(1)$$E=\frac{R_{0}^{6}}{R_{0}^{6}+R^{6}}$$
with $R_{0}$ being the Förster distance. The Förster distance is the distance at which the FRET efficiency is equal to 50 % (FRET and donor fluorescence are equally probable). The Förster distance is a FRET-pair-specific parameter and can be determined as followed.
(2)$$R_{0}^{6}=\frac{9\ln10\kappa^{2}\phi_{D}J\left(\lambda \right)}{128\pi^{5}n^{4}N_{AV}}$$

Here, $\kappa^{2}$ is the dipole orientation factor, $\phi_{D}$ the donor’s quantum yield, n is the refractive index of the surrounding media, $N_{AV}$ is Avogadro’s number and J(λ) is the overlap integral which represents the spectral overlap between the donor’s emission spectrum and the acceptor’s absorption spectrum (Appendix A). The spectral overlap integral is as follows,
(3)$$J\left(\lambda \right)=\frac{\int F_{D}\left(\lambda \right)\epsilon_{A}\left(\lambda \right)\lambda^{4}d\lambda }{\int F_{D}\left(\lambda \right)d\lambda }$$
where λ is the wavelength, $F_{D}\left(\lambda \right)$ is the donor’s emission spectrum and $\epsilon_{A}\left(\lambda \right)$ is the acceptor’s extinction coefficient spectrum. The Förster distance for FAM and Cy3 has been found to be 6.7 nm (see Appendix A). The FRET efficiency has been determined based on the donor’s fluorescence decay time as follows,
(4)$$E=1-\frac{\tau_{DA}}{\tau_{D}}$$
with $\tau_{DA}$ being the donor’s fluorescence decay time in presence of the acceptor and $\tau_{D}$ the fluorescence decay time in absence of the acceptor molecule ($\tau_{D}$ = 4.3 ns). The FAM fluorescence decay curves in presence of different salts have been measured using time-correlated single photon counting (TCSPC, see experimental section) and have been fitted tri-exponentially (three different decay time components, each decay time component corresponds to $\tau_{DA}$). Each decay time component can be attributed to a certain donor-acceptor distance. The FAM-Cy3 distances can be determined based on Equation (5) [47,48].
(5)$$R=R_{0}\sqrt[{6}]{\frac{1}{E}-1}$$

Based on the experimental FAM-Cy3-distances (corresponding to 5′-3′-distances) certain G-quadruplex structures can be assigned in presence of different salts. An overview of possible 5′-3′-distances in different G-quadruplex structures is shown in Figure 1. The 5′-3′-end can be at distances of approximately 4.1 nm, 5.8 nm, 4.6 nm or 6.1 nm (a linker length of 0.7 nm between fluorophore and DNA has been considered) depending on the position of 5′- or 3′-end in the G-quadruplex resulting from the specific loop and strand orientations [12,13].

In the presence of KCl, three different decay time components have been found with $\tau_{DA,1}$ = 0.24 ns, $\tau_{DA,2}\;$= 0.66 ns, $\tau_{DA,3}$ = 4.01 ns (fluorescence decay curve in presence of KCl is shown in Figure 2A). The third decay time component corresponds to a distance of 10.4 nm which can be attributed to the unfolded G-quadruplex (R (FAM-Cy3) = 26 nb ≈ 8.6 nm, when determined based on the structural properties of double-stranded DNA) [49]. The first and second decay time components correspond to distances of 4.2 nm and 5.0 nm, respectively, and can be ascribed to two different G-quadruplex structures. For a distance of 4.2 nm, a G-quadruplex structure, where the 5′- and 3′-end are located next to each other on the same G-tetrad (Figure 1 red), can be assigned (nominal distance of 4.1 nm). A distance of 5.0 nm can be ascribed to a G-quadruplex where the 5′- and 3′-end are located on different G-tetrads and on the same side of the G-quadruplex oriented diagonally to each other (nominal distance 4.6 nm; see Figure 1 green). This is in tandem with the CD results obtained in Figure 1A, which indicates the “hybrid-type” G-quadruplex structure.

For NaCl, the following decay time components have been found: $\tau_{DA,1}$ = 0.27 ns, $\tau_{DA,2}\;$= 0.78 ns and $\tau_{DA,3}$ = 3.94 ns. The FAM fluorescence decay curve in presence of NaCl is shown in Figure 2B. These decay time components correspond to the following distances *R*_1_ = 4.3 nm, *R*_2_ = 5.2 nm and *R*_3_ = 10 nm, respectively. Again, the distance based on the third decay time component belongs to the unfolded G-quadruplex structure. A distance of 4.3 nm can be attributed to a G-quadruplex structure where the 5′- and 3′-end are located next to each other on the same G-tetrad (Figure 1, red) and can be assigned to “chair-type” (nominal distance of 4.1 nm) or it could be that the 5′- and 3′-end are located on different G-tetrads and on the same side of the G-quadruplex structure oriented diagonally to each other (theoretical distance: 4.6 nm calculated considering the spacer distance of 0.7 nm between the fluorophore and DNA base sequence, see Figure 1 green). On the other hand, when the 5′- end 3′-end are located on the same G-tetrad and are oriented diagonally to each other (different side of the G-quadruplex, see Figure 1 yellow) a distance of 5.8 nm is expected which corresponds well with the determined distance of 5.2 nm. This indicates the formation of a “hybrid-type” or a mixture of parallel and anti-parallel G-quadruplexes supported by CD analysis.

The FAM fluorescence decay curves in presence of MgCl_2_ are depicted in Figure 2C. With a tri-exponential fit the fluorescence decay time components have been found to be $\tau_{DA,1}$ = 0.24 ns, $\tau_{DA,2}\;$= 1.07 ns, $\tau_{DA,3}$ = 3.78 ns. The third decay component corresponds to a distance of 9.3 nm which can be assigned to the unfolded G-quadruplex. For the first decay time component, a distance of 4.2 nm has been determined which can be ascribed to a G-quadruplex structure where the 5′- and 3′-end are located on different G-tetrads and oriented diagonally to each other (expected distance: 4.6 nm, Figure 1 green). The second decay time component corresponds to a distance of 5.6 nm which—along with the CD results discussed below—can be assigned to a G-quadruplex structure where the 5′- and 3′-end are located on different G-tetrads and oriented diagonally to each other (different sides of the G-quadruplex, nominal distance: 6.0 nm, Figure 1 blue) termed as new-hybrid.

For CaCl_2_ addition, similar values compared to MgCl_2_ addition have been found. The FAM fluorescence decay curves in presence of CaCl_2_ are shown in Figure 2D. The third decay time component ($\tau_{DA,3}$ = 3.80 ns) corresponds to a distance of 9.4 nm which belongs to the unfolded G-quadruplex structure. A distance of 4.1 nm has been determined for the first decay time component ($\tau_{DA,1}$ = 0.24 ns). Again, this can be ascribed to a G-quadruplex structure where the 5′- and 3′-end are located on different G-tetrads and oriented diagonally to each other (expected distance: 4.6 nm, Figure 1 green). Finally, a distance of 5.5 nm ($\tau_{DA,2}\;$= 1.07 ns) can be attributed to a G-quadruplex structure where the 5′- and 3′-end are located on different G-tetrads and oriented diagonally to each other (theoretical distance: 6.0 nm, Figure 1 blue) which we assign as new-hybrid structure. The above assignment for MgCl_2_ and CaCl_2_ holds true based on the results from CD spectra which indicate the parallel conformation of the structure.

## 3. Methods and Materials

### 3.1. Materials and Chemicals

The telomeric DNA strands (RevHumTel and HumTel), unmodified and modified with organic dyes, have been acquired from Metabion International AG (Planegg, Germany) (HPLC purified, dissolved in water). All DNA strands have been used as delivered without further treatment. Magnesium chloride (MgCl_2_, ≥98 %), sodium chloride (NaCl) and Tris acetate-EDTA buffer (TAE buffer, 10× concentrated) have been acquired from Sigma Aldrich (Hamburg, Germany). The diluted TAE buffer (1× concentrated, in ultrapure water (Merck KGaA, Darmstadt, Germany)) (pH = 8.2) contains 40 mM Tris-acetate and 1 mM EDTA. Calcium chloride (CaCl_2_, ≥98 %) has been purchased from Carl Roth (Karlsruhe, Germany) and potassium chloride (KCl) from VWR Prolabo (Darmstadt, Germany). The salt solutions have been prepared in ultrapure Millipore water.

### 3.2. CD Spectroscopy

CD spectroscopy has been performed on a JASCO J-815 CD spectrometer (JASCO Labor- und Datentechnik GmbH, Pfungstadt, Germany) with 1 mm quartz cuvettes from Hellma Analytics. For this, the DNA strands (unmodified RevHumTel (5′-TT (GGG ATT)_4_) and HumTel (5′-TT (GGG TTA)_3_ GGG TTT), c = 100 µM) have been diluted in TAE buffer (1× concentrated, incl. the respective salt concentration indicated below) to a final DNA concentration of 8.6 µM. KCl, NaCl, MgCl_2_ or CaCl_2_ (c = 200 mM, 2 M) have been used to induce the G-quadruplex formation (incubation for 15 min at room temperature, Appendix A shows that annealing at 40 °C yields the same result). Each sample has been prepared separately for each salt concentration ranging from 0 to 500 mM (c = (2, 10, 20, 200, 500) mM). The pure TAE buffer has been measured individually and subtracted from the sample’s CD spectrum. For the CD experiment of an unmodified RevHumTel DNA strand in the presence of a mixture of four different salts corresponding to physiological conditions (Na^+^: 143 mM; K^+^: 5 mM; Ca^2+^: 2.5 mM; Mg^2+^: 1 mM), a final concentration of 8.6 μL in 1X TAE is maintained. A blank buffer was prepared for each reaction, measured and subtracted from the sample CD spectrum. The CD spectra have been recorded from 200 to 320 nm with the following settings: digital integration time: 4 s, sensitivity: standard, spectral bandwidth: 1 nm, data pitch: 1 nm, start mode: immediately, scanning mode: continuous, scanning speed: 50 nm/min, accumulation: 3.

### 3.3. Melting Curves

The RevHumTel DNA diluted in 1X TAE buffer, with the final concentration of 3 μM, is considered. Four different batches of four different cations (KCl, NaCl, MgCl_2_, CaCl_2_) with a final concentration of 200 mM each are prepared. The melting temperature analysis has been recorded on a SPECORD 200 UV/Vis spectrometer (Analytik Jena AG, Jena, Germany) in a standard quartz cuvette. The temperature of the cell holder was increased from 4 °C to 90 °C at a rate of 1 °C/min (integration time: 0.1 s) and the UV-absorbance is collected from 250 to 320 nm.

### 3.4. Time-Correlated Single Photon Counting

For time-correlated single photon counting (TCSPC) measurements, the DNA (RevHumTel modified with FAM and Cy3, 5′-Cy3-TT (GGG ATT)_4_-FAM, c = 100 µM) has been diluted in a TAE (1×) buffer to a concentration of 5 nM. After the addition of salts to induce the G-quadruplex formation (KCl (c = 195 mM), NaCl (c = 490 mM), MgCl_2_ (c = 30 mM) and CaCl_2_ (c = 2 mM)) the samples have been incubated for 15 min at room temperature and measured afterwards. TCSPC measurements have been performed on a FLS920 fluorescence spectrophotometer from Edinburgh Instruments Ltd. with the F900 software (Edinburgh Instruments Ltd, Livingston, UK.) using 3 mm quartz cuvettes (Hellma Analytics). The samples have been measured in a 90° setup. As an excitation source, a supercontinuum white light source SC-400-PP from Fianium/NKT Photonics A/S (0.5–20 MHz, 400 nm < λ < 24,000 nm, pulse width: ca. 30 ps) and, as a detector, a multi-channel-plate ELDY EM1-132/300 from Europhoton GmbH have been used. The excitation wavelength has been set to (490 ± 1) nm and the emission wavelength to (520 ± 1) nm. The fluorescence decay curves (intensity-time-function *I*(*t*)) have been fitted with tri-exponential functions (Equation (6)) depending on the sample using the FAST software (Edinburgh Instruments, Livingston, UK).
(6)$$I\left(t\right)=\overset{n}{\underset{i=1}{\sum }}A_{i}e^{-\frac{t}{\tau_{i}}}$$

Here, $\tau_{i}$ is the decay time component and $A_{i}$ is the amplitude characteristic for each decay time component.

## 4. Conclusions

After analyzing the CD spectra and fluorescence decay curves of RevHumTel in the presence of different salts, we can predict the specific G-quadruplex structures which are folded in different environments. Please note that the donor-acceptor-distances determined in this work are error prone. Since the value for $R_{0}$ depends on a variety of parameters and not all of them are easily accessible (e.g., $\kappa^{2}$), the $R_{0}$ calculated here are not necessarily exact. Nevertheless, by combining CD spectroscopy with FRET measurements we can get closer insights into the most probable G-quadruplex structures formed in the presence of different salts. Based on the CD spectra the strand and loop orientations are accessible while the FRET experiments yield the locations of 5′- and 3′-end in the G-quadruplex. Furthermore, based on the FRET measurements it is evident that at least two different G-quadruplex structures are present for all salts. The different possible G-quadruplex structures are shown in Figure 3. As discussed above the CD spectrum in presence of KCl indicates the formation of a “hybrid-type” G-quadruplex structure which is also confirmed by a 5′-3′-distance of 5.0 nm (Figure 3A, right). A distance of 4.2 nm can be ascribed to an anti-parallel “chair-type” G-quadruplex structure (Figure 3A, left) which can also be confirmed in the CD spectrum (positive peak at 288 nm, negative peak at 240 nm). For NaCl, the CD spectrum is relatively ambiguous (“hybrid-type” or mixture of parallel/anti-parallel G-quadruplex structures). Together with the FRET experiments three different G-quadruplex structures in presence of NaCl are conceivable. A 5′-3′-distance of 5.2 nm can be attributed to an anti-parallel G-quadruplex structure (“basket-type”, Figure 3B, right). A distance of 4.3 nm could be an indication of an antiparallel chair-type conformation (Figure 3A, left), the formation of a parallel G-quadruplex structure (“propeller-type”, Figure 3B, left) or a “hybrid-type” G-quadruplex (two lateral and one external loop, Figure 3B, middle). For MgCl_2_ and CaCl_2_, the CD spectra indicate the formation of the same G-quadruplex structures which can also be confirmed by FRET measurements (determination of similar distances, *R*_1_ = 4.2 nm/4.1 nm, *R*_2_ = 5.6 nm/5.5 nm). Furthermore, the CD spectra indicate the formation of only parallel G-quadruplex structures (the distance of 4.2 nm/4.1 nm has to belong to a G-quadruplex where the 5′- and 3′-end are located on different G-tetrads and oriented diagonally to each other). Since two different distances in the FRET experiments have been found for the folded G-quadruplex structure the parallel G-quadruplex structures are based on different strand routings as shown in Figure 3C, D for MgCl_2_ and CaCl_2_, respectively. The G-quadruplex structures and conformation attained by RevHumTel in presence of different salts has been summarized in Table 1 based on theoretical and experimental results obtained in this study. In future research, the combination of FRET and CD measurements could be used to explore the folding of other G quadruplex structures and the method can be extended to other biologically relevant ions.

## Data Availability

The data presented in this study are available on request from the corresponding author.

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
