# Peer review of "Influence of Different Salts on the G-Quadruplex Structure Formed from the Reversed Human Telomeric DNA Sequence"

_ijms, 2022, doi:10.3390/ijms232012206_

Round 1

Reviewer 1 Report

The work done by Olejko et al investigated the effect of different salts on the G4 formation from RevHumTel by CD and FRET. By combinational analysis of the CD and FRET measurements, the authors worked out a method to reveal the conformation, orientation and combination of G4s in different salt solutions more precisely compared to only take one technique. This method is insightful and provided a different strategy to confirm G4 structures. Overall, this work is well designed and performed, below are additional comments that can help enhance the work further:

1. Test the CD and FRET of RevHumTel in the physiological salt conditions which is a mix of different cations (K+, Na+, Mg2+ and Ca2+ would be enough);

2. “Please note that MgCl2 and CaCl2 are able to quench fluorescence at high concentrations”: give the reason why the fluorescence quenched in the presence of high concentrations of MgCl2 and CaCl2, add citations;

3. “Please note that the donor-acceptor-distances determined in this work are error prone”: regarding the “error prone”, provide the estimation of the margin of error to indicate the accuracy of the showed structure conformation in Fig. 3;

4. “Based on the FRET measurements it is evident that at least two different G-quadruplex structures are present for all salts”: could you do melting experiment (either CD or UV melting) to measure the Tm of G4 with different cations? Do the melting curves also exhibit multiple melting stages?

Reviewer 2 Report

The manuscript by Olejko et al. examines the approximate structure formed by the reversed human sequence G-quadruplex (GQ) in different salt types and concentrations, using circular dichroism (CD) and Förster resonance energy transfer (FRET).  The authors interpret the CD and FRET results to suggest that the in the presence of these different the molecule adapts different types of GQs, and that the ions encourage formation of multiple GQ species.

The paper a good addition to the greater literature and requires only minor changes. These are outlined below.

Experimental design

-        What is the rationale for why the strand concentration is 8.6 μM?  

-        The salt solutions were prepared in water and the DNA in buffer. Would the addition of the salt solutions have decreased the concentration of the DNA and buffer? 

-        It appears samples were not annealed after adding the salt solutions to the DNA. Is this true?  Would the results have been different had the DNA been annealed? It may be possible there are kinetically trapped states when salts are added at room temperature and samples equilibrated for 15 mins.  

Results 

-        Are all the structure formed at the different salt concentrations intramolecular GQ; that is, are they monomolecular? Is there evidence for monomolecularity? 

-        In line 142, the Authors state that the salt concentration does not influence the CD spectrum, but this is not necessarily true, as the shape changes slightly in the case of KCl, MgCl2, and CaCl2. Suggest rephrasing this sentence 

-        In Fig 1D, from 200 to 500 mM CaCl2, the intensity of the CD signal decreases. The Authors rationalize this as the increase in salt concentration is causing destabilization of the structure.  Can the Authors suggest a possible mechanism for this? Is it possible something else is occurring, rather than the destabilization of the structure? 

-        It would benefit the reader to know if there are next steps for this research, or what more can be explored

Basic reporting

-        In some instances, the Authors could be more clear in their writing.  For example, in line 29, the sentence should be rephrased 

-        The results appear valid. They are written a way that is easy to follow, and the figures are nice 

For some of the figures, consider the colour scheme. For example, Figure 1C, the blues are difficult for this Reviewer to distinguish. 

-         

Round 2

Reviewer 1 Report

Accept